# Etoposide as a Key Therapeutic Agent in Lung Cancer: Mechanisms, Efficacy, and Emerging Strategies

**DOI:** 10.3390/ijms26020796

**Published:** 2025-01-18

**Authors:** Jung Yoon Jang, Donghwan Kim, Eunok Im, Nam Deuk Kim

**Affiliations:** 1Department of Pharmacy, College of Pharmacy, Research Institute for Drug Development, Pusan National University, Busan 46241, Republic of Korea; jungyoon486@pusan.ac.kr; 2Functional Food Materials Research Group, Korea Food Research Institute, Wanju-gun 55365, Jeollabuk-do, Republic of Korea; kimd@kfri.re.kr

**Keywords:** topoisomerase II inhibitors, etoposide, lung cancer, anticancer

## Abstract

Topoisomerase II inhibitors, particularly etoposide, have long been integral to the treatment of lung cancer, especially small cell lung cancer. This review comprehensively examines the mechanisms of action of etoposide, its clinical efficacy, and its role in current lung cancer treatment regimens. Etoposide exerts its anticancer effects by inducing DNA strand breaks through the inhibition of topoisomerase II, leading to cancer cell apoptosis. Despite their widespread use, challenges such as drug resistance, toxicity, and limited efficacy in non-small cell lung cancer have spurred ongoing research on combination therapies and novel drug formulations. Emerging therapeutic strategies include the integration of etoposide with immunotherapy, targeted therapies, and novel drug delivery systems aimed at enhancing the therapeutic window and overcoming drug resistance. This article aims to inform the development of more effective treatment strategies by providing a critical overview of the clinical applications of etoposide and exploring future directions for lung cancer therapy.

## 1. Introduction

Cancer remains a leading cause of mortality worldwide, with an estimated 20 million new cases and nearly 9.7 million cancer-related deaths reported in 2022. Among these, lung cancer is a major contributor, accounting for approximately 12.4% (ranked first) of all cancer cases and 18.7% (ranked first) of cancer-related deaths [1]. This high mortality rate is largely attributable to the aggressive nature of lung cancer, which is often diagnosed at advanced stages when treatment options are limited and less effective [2]. Lung cancer is typically categorized into two primary histological types: non-small cell lung cancer (NSCLC), which accounts for approximately 85% of all lung cancer cases, and small cell lung cancer (SCLC), comprising the remaining 15% [3]. Each subtype presents distinct challenges and prognoses, complicating treatment efforts [2].

The high prevalence and mortality of lung cancer have driven extensive research into more effective therapeutic options [4]. The latest treatment methods for lung cancer include traditional treatments such as chemotherapy, radiation therapy, and surgery, as well as newer approaches such as targeted therapy, immunotherapy, gene therapy, and personalized medicine [5]. Chemotherapy for lung cancer began in the late 1940s and has been in use for over 70 years, and it remains one of the most useful treatment methods to this day [6]. The ten most commonly used chemotherapeutic drugs for lung cancer are cisplatin, carboplatin, paclitaxel, docetaxel, etoposide, vincristine, methotrexate, pemetrexed vincristine, mercaptopurine, and chlorambucil [7]. These drugs are often used in various combinations, depending on the type of lung cancer and the patient’s condition [8].

Among chemotherapeutic options, topoisomerase II inhibitors, particularly etoposide, have exhibited promise for treating both NSCLC and SCLC [9,10]. Etoposide, a semi-synthetic derivative of podophyllotoxin, exerts anticancer effects by inhibiting topoisomerase II, an enzyme essential for DNA replication and cell division [11]. By stabilizing the enzyme–DNA complex, etoposide induces DNA strand breaks, promoting apoptosis in rapidly dividing cancer cells [12]. Due to its ability to selectively target tumor cells with high proliferative indices, etoposide has become a cornerstone of lung cancer treatment regimens, particularly in combination therapies [13].

Despite its demonstrated efficacy, challenges such as drug resistance and adverse side effects significantly undermine patient outcomes and diminish quality of life [14]. These limitations underscore the need for the continued exploration of combination therapies and novel formulations to enhance etoposide therapeutic potential while minimizing toxicity [15,16]. This review provides a comprehensive examination of the role of the topoisomerase II inhibitor, etoposide, in lung cancer treatment and presents the current clinical status of etoposide in lung cancer therapy. We performed searches in the PubMed, Google Scholar, and ClinicalTrials.gov databases using the keywords “etoposide” and its brand name “VP-16.” The search covered both preclinical and clinical studies published up to December 2024, with an emphasis on full-text articles in English. Clinical study data were retrieved from ClinicalTrials.gov using the keywords “lung,” “etoposide,” and “VP-16,” concentrating on studies that reported results from 2018 onward, including the most recent data available.

## 2. Mechanism of Action of Etoposide

Etoposide, an effective chemotherapeutic agent, was approved by the U.S. Food and Drug Administration (FDA) in 1983 [17]. It is also included in the World Health Organization’s (WHO) Essential Medicines List (EML) [18] (Figure 1).

The WHO EML designates essential medicines that are crucial for meeting major healthcare needs [19]. Etoposide is included on the EML as an antineoplastic agent due to its vital role in treating various cancers, including lung cancer [18]. It is listed under chemotherapeutic agents, recognized for its effectiveness, safety, and cost-efficiency in resource-limited environments [20]. Other essential medicines in this category, such as doxorubicin, cyclophosphamide, and cisplatin, are also commonly used in cancer treatment, similar to etoposide [18].

Initially approved for use in SCLC, etoposide is widely used in combination with other agents as a first-line chemotherapy regimen for this aggressive cancer type [21]. Beyond lung cancer, etoposide has been approved for treating malignancies, such as testicular cancer [22], leukemia [23], lymphoma [24], neuroblastoma [25], and ovarian cancer [26]. It is also used to treat hemophagocytic lymphohistiocytosis [27]. Etoposide is also employed off-label or as part of combination chemotherapy regimens in the management of other cancers, such as brain tumors (e.g., glioma), osteosarcoma [28], and gastric cancer [29] (Figure 2).

Etoposide exerts its anticancer effects by targeting topoisomerase II, an enzyme essential for managing DNA topology during replication and transcription [30]. Topoisomerase II facilitates the transient cleavage of both DNA strands, allowing the resolution of supercoils and entanglements that occur during cellular processes [31]. Once DNA relaxation or decatenation is achieved, topoisomerase II releases the DNA strands, thereby restoring genomic integrity [32]. Etoposide disrupts this process by stabilizing the transient cleavage complex between topoisomerase II and DNA, thereby preventing the re-ligation step [33] (Figure 3).

This results in the accumulation of double-strand breaks (DSBs), leading to DNA damage that cells cannot readily repair [34]. Accumulated DSBs trigger cellular responses, such as cell cycle arrest, apoptosis, and mitotic catastrophe, depending on the extent of damage and the cell’s repair capacity [35]. Furthermore, etoposide exerts cytotoxic effects most effectively during the S and G2 phases of the cell cycle, when DNA replication and repair processes are actively engaged. This phase-specific action enhances selectivity for rapidly proliferating cancer cells [36]. Importantly, by inducing DNA fragmentation and promoting apoptosis, etoposide is particularly potent against lung cancer cells that often exhibit high mitotic rates [37].

Etoposide, a topoisomerase II inhibitor, mainly induces apoptosis through DNA damage and cell cycle arrest, but it can also initiate various other forms of cell death, demonstrating its intricate mechanisms of action [38,39,40,41,42,43,44]. These include the induction of necrosis via p53-mediated antiapoptotic pathways [38]; necroptosis regulated by the receptor-interacting serine/threonine-protein kinase (RIPK)1, RIPK3, and mixed lineage kinase domain-like protein (MLKL) pathways [39]; ferroptosis through reactive oxygen species (ROS) production and lipid peroxidation [40]; pyroptosis through caspase activation [41]; and parthanatos due to the hyperactivation of poly(ADP-ribose) polymerase (PARP)1 [42]. Additionally, etoposide simultaneously triggers autophagic cell death and apoptosis [43]. Etoposide can also trigger immunogenic cell death, enhancing anti-tumor immunity by releasing damage-associated molecular patterns [44].

## 3. Application of Etoposide in Lung Cancer Treatment

### 3.1. Lung Cancer Treatment

Lung cancer poses significant challenges for early detection and effective treatment. It is typically categorized into two main types: NSCLC, which accounts for approximately 85% of the cases, and SCLC, which is more aggressive but accounts for approximately 15% of the cases [45]. Due to its aggressive nature and high proliferative rate, SCLC is particularly difficult to treat and is linked to a poorer prognosis [46].

Treatment modalities have evolved significantly, primarily encompassing surgery, radiation therapy, and systemic therapies, including chemotherapy, targeted therapy, and immunotherapy [47]. For localized NSCLC, surgical resection remains the primary curative treatment with lobectomy or wedge resection preferred over pneumonectomy due to better outcomes and reduced morbidity [48]. Combination chemotherapy regimens, including cisplatin or carboplatin with etoposide, have demonstrated efficacy in both SCLC and NSCLC, particularly in patients with unresectable disease [49]. Recent advances in molecular profiling have facilitated the development of targeted therapies for NSCLC, including tyrosine kinase inhibitors (TKIs) designed for patients with specific mutations, such as those in the epidermal growth factor receptor (EGFR) or anaplastic lymphoma kinase (ALK). Immunotherapy has also exhibited significant promise for NSCLC, with immune checkpoint inhibitors (ICIs) (such as programmed death-1 (PD-1)/programmed death-ligand 1 (PD-L1) inhibitors) being part of the standard treatment for advanced NSCLC [50]. The emergence of immunotherapies, especially ICIs like pembrolizumab and nivolumab, has greatly broadened treatment possibilities for advanced lung cancer. These drugs work by blocking inhibitory signals in the immune system, enhancing the body’s anti-tumor response [51]. Investigational treatments aim to enhance therapeutic outcomes by combining etoposide with other innovative agents, such as PARP inhibitors or alternative topoisomerase inhibitors. These combinations exploit various cellular pathways, potentially addressing drug resistance and improving therapeutic efficacy [52].

### 3.2. Treatment Combination Therapies with Etoposide

Combination therapies have become integral to lung cancer treatment, often enhancing therapeutic efficacy by targeting multiple pathways involved in cancer progression [50]. Etoposide, a topoisomerase II inhibitor, exhibit significant synergistic effects when used in combination with other chemotherapeutic agents and targeted therapies [53]. This approach aims not only to maximize the cytotoxic effects on malignant cells, but also to potentially reduce resistance and mitigate the required dosage of each drug, and this thus helps minimize side effects [54].

#### 3.2.1. Etoposide and Platinum-Based Agents

Etoposide is frequently used in combination with platinum-based agents, such as cisplatin or carboplatin [55]. The etoposide and cisplatin (EP) regimen is the cornerstone of SCLC treatment, particularly for limited-stage SCLC (LS-SCLC) and extensive-stage SCLC (ES-SCLC) [56]. This regimen leverages the synergistic cytotoxic effects of cisplatin, a platinum-based agent that forms DNA crosslinks, and etoposide that inhibits topoisomerase II, leading to irreparable DNA damage and the subsequent apoptosis of rapidly proliferating cancer cells [57]. The typical EP regimen involves a cycle every 3–4 weeks, with etoposide administered intravenously for the first 3–5 days of the cycle, and cisplatin administered on the first day. Common dosing regimens include etoposide (100 mg/m^2^, days 1–3) and cisplatin (75 mg/m^2^, day 1). This cycle is repeated for 4–6 cycles, with rest periods between each cycle to allow for patient recovery and bone marrow replenishment [58]. The EP regimen exhibits high response rates, with clinical trials reporting response rates as high as 60–80% in patients with ES-SCLC [59].

The etoposide and carboplatin (EC) regimen is an alternative to EP that is particularly suited for elderly patients who may not tolerate cisplatin owing to renal impairment or frailty [60]. The comparatively milder toxicity profile and easier administration of carboplatin make it an attractive option for the first-line therapy of ES-SCLC [61]. Similarly to EP, the EC regimen involves a multi-day cycle that is repeated every 3–4 weeks. The standard dose consists of etoposide (100 mg/m^2^, days 1–3) with carboplatin dosed according to the area under the curve (AUC 5–6, day 1), which is calculated to adjust for renal function and patient weight. This process was typically repeated for up to six cycles [62]. Although carboplatin is less nephrotoxic than cisplatin, EC generally exhibits comparable response rates to EP, with a lower incidence of nephrotoxicity and ototoxicity [13,63].

#### 3.2.2. Etoposide and Concurrent Chemoradiotherapy (CCRT)

In LS-SCLC, combining etoposide with CCRT is standard practice. This combination enhances local tumor control and improves survival rates by exploiting the radiosensitizing properties of etoposide [60]. The EP regimen is typically paired with thoracic radiotherapy, starting either during or immediately after the first cycle of chemotherapy [64]. Radiation is typically administered at a dose of 45 gray (Gy) over 30 sessions or, alternatively, twice daily to enhance local control [65].

#### 3.2.3. Etoposide with Immunotherapy

Recent advancements in immunotherapy have led to the incorporation of ICIs such as atezolizumab and durvalumab with etoposide and carboplatin for ES-SCLC [66]. This combination is based on the premise that etoposide-induced DNA damage can increase tumor antigen presentation, thereby enhancing immune recognition [67]. Standard dosing involves the EC regimen alongside ICIs, with atezolizumab (1200 mg) or durvalumab (1500 mg) administered intravenously once every three weeks. ICIs are continued as maintenance therapy following the initial chemotherapy cycles, provided there is no disease progression [68]. Studies, such as the IMpower133 trial, have demonstrated that adding atezolizumab to the EC regimen significantly improves median overall survival and progression-free survival in patients with ES-SCLC [69].

#### 3.2.4. Etoposide with Targeted Therapies

Although less established, the combination of etoposide with targeted therapies is under investigation, particularly in patients with NSCLC harboring specific genetic mutations such as *EGFR* or *ALK* [70]. Etoposide may act synergistically with agents such as erlotinib (an EGFR inhibitor) in NSCLC by enhancing cellular susceptibility to targeted apoptosis [71]. There is no standardized dosing for these regimens, as studies remain in the early phases [72]. Typical strategies involve administering etoposide with the targeted agent in parallel cycles or sequentially, depending on tolerability [73]. Although preclinical data suggest its promise, its clinical efficacy and safety remain unproven [50]. Research on targeted combination therapies with etoposide is ongoing, with trials focusing on patients’ genetic profiles to optimize the efficacy and reduce resistance [74].

## 4. Clinical Status of Etoposide in the Treatment of Lung Cancer

Currently, many clinical trials focused on etoposide for lung cancer treatment are in progress. The clinical trial results reported from 2018 to the most recently available data are summarized in Table 1.

NCT03041311 assessed the effects of trilaciclib administration before carboplatin, etoposide, and atezolizumab (E/P/A) in patients with newly diagnosed ES-SCLC [80,81]. Trilaciclib, an intravenous cyclin-dependent kinase 4/6 inhibitor, was designed to protect hematopoietic stem and progenitor cells and immune functions from chemotherapy-induced damage (myelopreservation) [115]. Trilaciclib significantly reduced the mean duration of SN (absolute neutrophil count) in cycle 1 (0 vs. 4 days; *p* < 0.0001) and the incidence of severe neutropenia (1.9% vs. 49.1%; *p* < 0.0001) compared to that in response to placebo. Additional benefits included improvements in red blood cell and platelet counts, enhanced health-related quality of life (HRQoL), and fewer grade ≥3 adverse events, primarily due to reduced high-grade hematologic toxicity. The anti-tumor efficacy outcomes were similar between groups. Notably, trilaciclib promoted greater peripheral T-cell clone expansion (*p* = 0.019), with a significant increase observed in patients who responded to E/P/A (*p* = 0.002). These findings demonstrate that trilaciclib enhances patient outcomes by mitigating myelosuppression, improving safety profiles, and preserving HRQoL during ES-SCLC treatment [80,81].

NCT03066778 evaluated pembrolizumab, an anti-programmed death-(PD)-1 immune checkpoint inhibitor that enhances T-cell-mediated immune responses against tumors by blocking the interaction between PD-1 and its ligands (PD-L1 and PD-L2), combined with etoposide and platinum versus placebo plus etoposide and platinum as a first-line treatment for ES-SCLC. Among 453 patients, those receiving pembrolizumab plus etoposide and platinum exhibited significantly improved progression-free survival (PFS), with 12-month PFS rates of 13.6% compared to 3.1% in the placebo group. Although overall survival (OS) was prolonged, the predefined significance threshold was not met. The objective response rate (ORR) was higher in the pembrolizumab group (70.6% vs. 61.8%), with more durable responses at 12 months (19.3% vs. 3.3%). Adverse events of grade 3–4 occurred in 76.7% of pembrolizumab-treated patients versus 74.9% of placebo patients, with no unexpected toxicities observed. These findings highlight the potential of pembrolizumab for enhancing the outcomes of ES-SCLC [88,89].

## 5. Toxicity, Side Effects, and Resistance Mechanisms of Etoposide

Etoposide, a chemotherapeutic agent that inhibits topoisomerase II, is associated with various adverse effects that complicate its application in lung cancer treatment [9]. Primary toxicities include myelosuppression, gastrointestinal disturbances, and hypersensitivity reactions [116]. Myelosuppression, especially severe neutropenia, represents a significant concern, impacting approximately 25% to 50% of patients and markedly increasing their risk of infection [117]. Close monitoring is essential and the use of granulocyte colony-stimulating factors may be necessary for high-risk patients [118]. Additionally, gastrointestinal side effects such as nausea and vomiting can detrimentally affect patients’ quality of life and adherence to treatment regimens [119]. Although rare, hypersensitivity reactions may occur during etoposide administration, emphasizing the need for careful observation [120]. Long-term risks include alopecia and the potential to develop secondary malignancies, particularly acute myeloid leukemia and myelodysplastic syndromes [121]. Given its toxicity profile, etoposide is often used in combination therapies [122]. These combinations aim to enhance therapeutic efficacy while minimizing the adverse effects associated with high doses of etoposide, thereby enhancing overall patient outcomes and treatment tolerability [123].

The side effects associated with etoposide highlight the need for combination therapies that leverage its therapeutic benefits while minimizing toxicity. This approach ultimately optimizes treatment strategies for patients with lung cancer [124].

Although etoposide is effective in treating lung cancer, its efficacy is limited by cellular resistance mechanisms, such as alterations in topoisomerase II, increased drug removal through ATP-binding cassette (ABC) transporters, and enhanced DNA repair processes [125]. These mechanisms reduce the drug’s efficacy and contribute to treatment resistance [126]. Strategies to overcome this resistance include using ABC transporter inhibitors [127], developing novel topoisomerase II inhibitors [128], and combining etoposide with DNA repair inhibitors like PARP inhibitors [129]. Additionally, nanoparticle-based drug delivery systems aim to improve targeting and reduce off-target effects [130]. In general, progress in combination therapies, new formulations, and strategies to overcome resistance provide optimism for improving etoposide’s effectiveness while minimizing side effects [131].

## 6. Conclusions

Etoposide remains a cornerstone of lung cancer treatment, owing to its efficacy as a topoisomerase II inhibitor. Its established role in both NSCLC and SCLC is reinforced by its efficacy in combination therapies, including platinum-based agents, CCRT, immunotherapy, and targeted treatments, that improve patient outcomes. However, its clinical utility is challenged by toxicity, drug resistance, and narrow therapeutic index, necessitating careful management and continued research. To overcome these challenges, promising approaches are being explored, including new formulations, optimized dosing strategies, and the integration of emerging treatments. Looking ahead, the clinical use of etoposide is expected to progress through advancements in drug delivery technologies, such as nanoparticle-based formulations, which could improve its therapeutic effectiveness by enhancing bioavailability and reducing off-target side effects. Furthermore, combining etoposide with next-generation precision medicine techniques, such as biomarker-guided treatment approaches, provides opportunities to customize therapies for individual patients, increasing efficacy while minimizing adverse effects. In addition, the development of combination therapies with novel treatments, such as PARP inhibitors, new ICIs, and anti-angiogenic agents, offers promising prospects for overcoming resistance mechanisms and expanding the therapeutic potential of etoposide. In conclusion, while etoposide retains its therapeutic significance, its future lies in modern combination strategies and precision medicine to maximize efficacy and improve patient outcomes in lung cancer management.

## Figures and Tables

**Figure 1 ijms-26-00796-f001:**
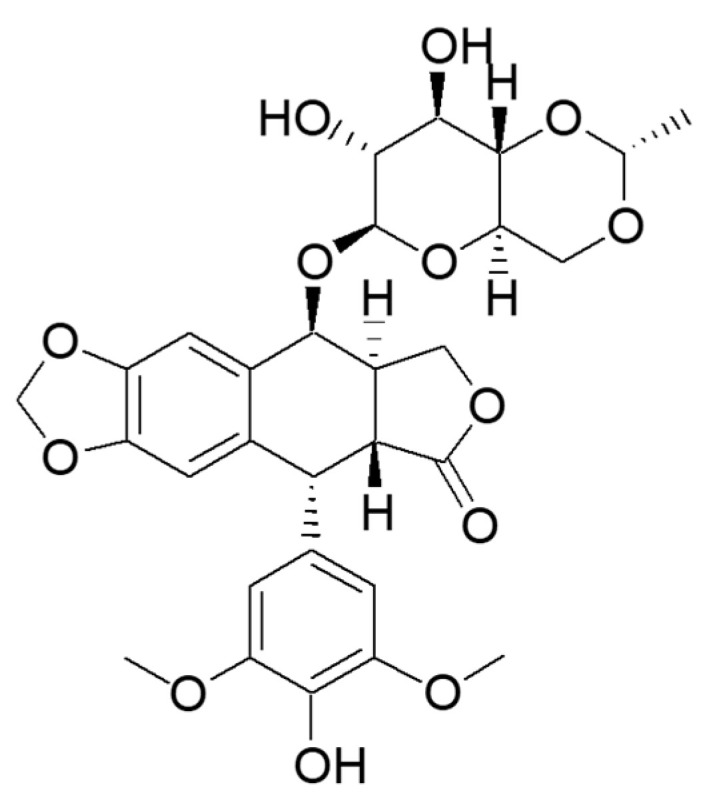
Structure of etoposide.

**Figure 2 ijms-26-00796-f002:**
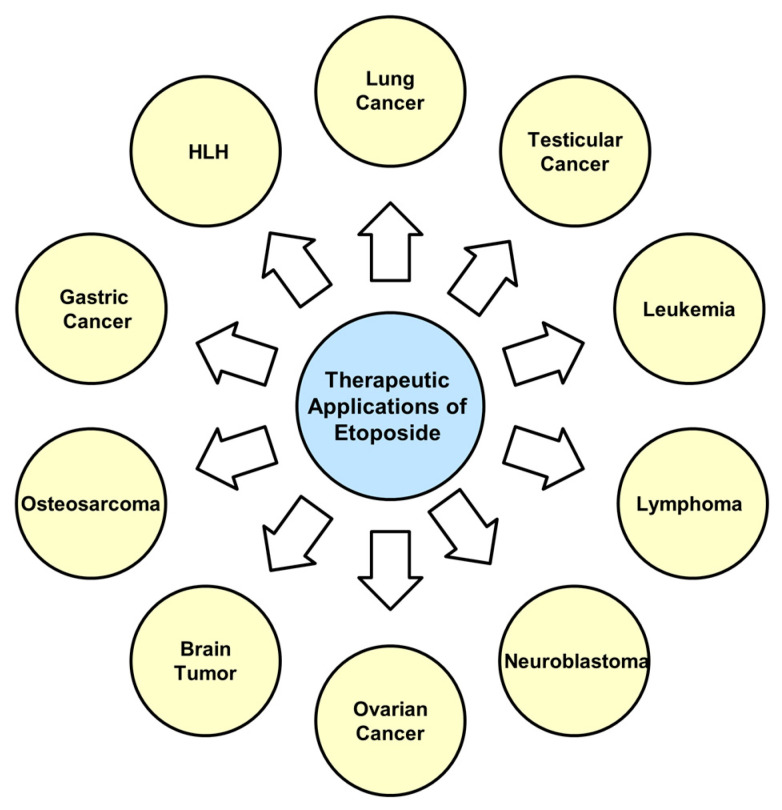
Therapeutic applications of etoposide. HLH, hemophagocytic lymphohistiocytosis.

**Figure 3 ijms-26-00796-f003:**
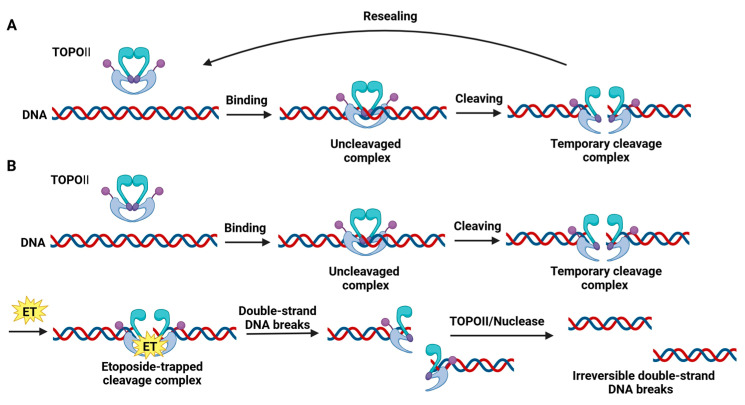
Etoposide exerts its mechanism of action by targeting and disrupting the function of topoisomerase II (TOPII). (**A**) Normally, TOPII facilitates the creation and resealing of double-strand breaks (DSBs) through a three-step process: binding to DNA, cleaving the double-stranded DNA, and resealing the transiently cleaved DNA; (**B**) etoposide inhibits TOPII by stabilizing the TOPII-DNA cleavage complex (TOPIIcc). When the trapped TOPIIcc cannot be efficiently resolved, the TOPII enzyme or nucleases remove TOPII from the complex, resulting in persistent and irreversible double-strand DNA breaks. Created in BioRender. Jang, J. created this figure in 2024, details can be found at this link https://BioRender.com/y13x678 (accessed on 23 December 2024). ET, etoposide.

**Table 1 ijms-26-00796-t001:** Clinical status of etoposide in lung cancer.

Interventions	Conditions	Study Title	Status	Phase	Refs.
Carboplatin, Etoposide, MPDL3280A	SCLC	Carboplatin plus etoposide with or without MPDL3280A in untreated ES-SCLC	Terminated	Phase 1, Phase 2	[75]
Atezolizumab, Carboplatin, Entinostat, Etoposide	ES-SCLC, malignant solid neoplasm, metastatic malignant neoplasm in the brain	Testing the addition of an anticancer drug, entinostat, to the usual chemotherapy and immunotherapy treatment (atezolizumab, carboplatin and etoposide) for previously untreated aggressive lung cancer that has spread	Completed	Phase 1	[76,77]
Carboplatin, Etoposide, Atezolizumab	SCLC, brain metastases	Chemotherapy and aezolizumab for patients with ES-SCLC with untreated, asymptomatic brain metastases	Terminated	Phase 2	[78]
Carboplatin, Cisplatin, Etoposide, Nivolumab	ES-SCLC, recurrent lung small cell carcinoma	Cisplatin/carboplatin and etoposide with or without nivolumab in treating patients with extensive stage small cell lung cancer	Active, Not recruiting	Phase 2	[79]
Trilaciclib, Placebo, Carboplatin, Etoposide, Atezolizumab	SCLC	Carboplatin, etoposide, and atezolizumab with or without trilaciclib (G1T28), a CDK4/6 inhibitor, in extensive-stage SCLC	Terminated	Phase 2	[80,81,82]
Cisplatin, Durvalumab, Etoposide, Hypofractionated radiation Therapy, Pemetrexed	Locally advanced lung non-small cell carcinoma, Stage III lung cancer AJCC v8, Stage IIIA lung cancer AJCC v8, Stage IIIB lung cancer AJCC v8, Stage IIIC lung cancer AJCC v8	ADMIRAL trial: Adaptive mediastinal radiation with chemo-immunotherapy	Terminated	Phase 2	[83]
Carboplatin, Cediranib, Cediranib maleate, Cisplatin, Etoposide, Olaparib	ES-SCLC	Olaparib, cediranib maleate, and standard chemotherapy in treating patients with small cell lung cancer	Terminated	Phase 2	[84]
Venetoclax, Atezolizumab, Carboplatin, Etoposide	SCLC	A study evaluating the safety, tolerability, pharmacokinetics, and efficacy of venetoclax in combination with atezolizumab, carboplatin, and etoposide in participants with untreated ES-SCLC	Terminated	Phase 1	[85]
Durvalumab, Cisplatin, Carboplatin, Etoposide	ES-SCLC	Study of durvalumab in combination with platinum and etoposide for the first line treatment of patients with extensive-stage small cell lung cancer (LUMINANCE)	Active, not recruiting	Phase 3	[86]
Trilaciclib, Carboplatin, Etoposide, or Topotecan	ES-SCLC	Phase 3 study evaluating efficacy, safety and pharmacokinetics of trilaciclib in SCLC patients	Completed	Phase 3	[87]
Pembrolizumab, Normal saline solution, Carboplatin, Cisplatin, Etoposide	SCLC	A study of pembrolizumab (MK-3475) in combination with etoposide/platinum (cisplatin or carboplatin) for participants with extensive stage small cell lung cancer (MK-3475-604/KEYNOTE-604)	Completed	Phase 3	[88,89]
Durvalumab, Tremelimumab, Carboplatin, Cisplatin, Etoposide	ES-SCLC	Durvalumab ± tremelimumab in combination with platinum based chemotherapy in untreated ES-SCLC (CASPIAN)	Active, not recruiting	Phase 3	[90,91,92]
M7824, Placebo, Durvalumab, Etoposide, Pemetrexed, Carboplatin, Paclitaxel, Cisplatin, Intensity modulated radiation therapy	NSCLC	M7824 With CCRT in unresectable stage III NSCLC	Terminated	Phase 2	[93,94,95]
Pembrolizumab, Cisplatin, Carboplatin, Etoposide, Radiation therapy	SCLC	Study of pembrolizumab and chemotherapy with or without radiation in SCLC	Terminated	Phase 2	[96]
Atezolizumab (MPDL3280A), Carboplatin, Etoposide, Placebo	SCLC	A study of carboplatin plus etoposide with or without atezolizumab in participants with untreated ES -SCLC (IMpower133)	Completed	Phase 3	[69,97,98,99,100]
Tislelizumab, Carboplatin/ Cisplatin, Etoposide, Carboplatin/ Cisplatin, Etoposide	SCLC	Study of platinum plus etoposide with or without BGB-A317 in participants with untreated extensive-stage small cell lung cancer	Completed	Phase 3	[101,102]
Platinum-etoposide+ Anlotinib	ES-SCLC	Anlotinib plus platinum-etoposide in first-line of ES-SCLC	Unknown status	Phase 2	[103,104]
SHR-1316, Carboplatin, Etoposide, Placebo	ES-SCLC	Study of carboplatin plus etoposide with or without SHR-1316 in participants with untreated ES-SCLC	Unknown status	Phase 3	[105,106]
BMS-986012, Carboplatin, Etoposide, Nivolumab	ES-SCLC	A study of BMS-986012 in combination with carboplatin, etoposide, and nivolumab as first-line therapy in ES-SCLC	Active, not recruiting	Phase 2	[107,108]
CC-90011, Cisplatin, Carboplatin, Etoposide, Nivolumab	SCLC	A safety, tolerability and preliminary efficacy evaluation of CC-90011 given in combination with cisplatin and etoposide in subjects with first line, ES-SCLC	Completed	Phase 1	[109,110]
HLX10, Carboplatin and etoposide, Placebo	ES-SCLC	A randomized, double-blind, placebo controlled phase III study to investigate efficacy and safety of HLX10 + chemotherapy (carboplatin-etoposide) in patients with ES-SCLC	Unknown status	Phase 3	[111,112]
RRx-001 + eLOOP Device, Cisplain, Carboplatin, Etoposide	Carcinoma, small cell lung cancer	RRx-001 sequentially with a platinum doublet or a platinum doublet in third-line or beyond in patients with SCLC (REPLATINUM)	Terminated	Phase 3	[113,114]

CCRT, concurrent chemoradiotherapy; CDK, cyclin-dependent kinase; ES-SCLC, extensive-stage small-cell lung cancer; NSCLC, non-small cell lung cancer; SCLC, small-cell lung cancer.

## Data Availability

Data presented in this study are available upon request from the corresponding author.

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
