# Peer review of "Etoposide as a Key Therapeutic Agent in Lung Cancer: Mechanisms, Efficacy, and Emerging Strategies"

_ijms, 2025, doi:10.3390/ijms26020796_

Round 1
Reviewer 1 Report
Comments and Suggestions for Authors
The manuscript titled "Topoisomerase II Inhibitors in Lung Cancer: A Comprehensive Review of Etoposide's Role and Emerging Therapeutic Strategies" provides an exploration of etoposide specifically focusing on its application in lung cancer treatment. Despite the advent of newer targeted therapies and immunotherapies, etoposide continues to be a critical component of chemotherapy regimens, particularly in SCLC, where it remains one of the cornerstones of treatment. The paper is timely and relevant, as ongoing research continues to investigate ways to enhance etoposide’s therapeutic impact while minimizing adverse effects. The review is expected to be of significant value to clinicians, researchers, and healthcare professionals involved in lung cancer treatment, as it highlights the critical role etoposide still plays in the clinical management of the disease. Nevertheless, there are some minor issues that need to be resolved before publication.
Major comment:
I suggest rewriting the title to: “Etoposide as a Key Therapeutic Agent in Lung Cancer: Mechanisms, Efficacy, and Emerging Strategies” or “The Role of Etoposide in Lung Cancer Treatment: A Detailed Review of Current and Future Strategies” better emphasizing that the manuscript focuses on etoposide.
In Section 2, the authors should explore whether etoposide can induce various forms of cell death beyond apoptosis, including necrosis, necroptosis, ferroptosis, pyroptosis, parthanatos, entotic cell death, NETotic cell death, lysosome-dependent cell death, autophagy-dependent cell death, and immunogenic cell death. A comprehensive understanding of these complementary cell death pathways is crucial, as it may reveal alternative mechanisms by which etoposide exerts its cytotoxic effects and contribute to a more complete characterization of its therapeutic potential and side effects.
Authors should include a section discussing the mechanisms of cellular resistance to etoposide and the strategies employed to overcome this resistance. Etoposide resistance can significantly impact treatment efficacy, leading to therapeutic failure. A comprehensive understanding of the molecular mechanisms underlying resistance—such as the role of drug efflux pumps, alterations in drug targets, enhanced DNA repair pathways, and dysregulation of apoptosis—can elucidate the reasons why certain tumors exhibit diminished sensitivity to the drug over time. This knowledge is essential for the development of strategies to overcome resistance. Additionally, such insights may enable the design of personalized treatment approaches, wherein patients receive therapies tailored to the specific resistance mechanisms present in their tumors, thereby improving therapeutic outcomes and minimizing unnecessary side effects.
Minor comment:
Line 17: I suggest changing “breakages” into “breaks”
Line 91: space lacking in “TOPâ…¡facilitates”
Line 120-122: sentence “Etoposide, a topoisomerase II inhibitor, plays a pivotal role in these regimens by inducing DNA damage, leading to cancer cell apoptosis” should be removed.
Line 127: lack of full names for abbreviations: “such as PD-1/PD-L1 inhibitor”
Line 128: The sentence is relatively clear but could benefit from some refinement for greater precision and flow e.g. "The introduction of immunotherapies, particularly immune checkpoint inhibitors such as pembrolizumab and nivolumab, has significantly expanded treatment options for advanced lung cancer. These agents block inhibitory signals in the immune system, thereby boosting antitumor activity."
Line 139 and 232: “Etoposide, a potent topoisomerase II inhibitor/Etoposide, a potent chemotherapeutic agent that inhibits topoisomerase II ”information is mentioned for the third time in the manuscript and should be removed.
Line 143: “this can help…” can be replace with “thus help…”
Line 153, 173, 190: remove the bold.
Line 126: “immune checkpoint inhibitors” appear the first time in the text and is later shortened in line 177. Please use abbreviation (ICIs) the first time it appears in the text.
Line 205: Include full name for CDK abbreviation.
Line 245: I suggest replacing “bolster” with “boost” or “enhance”
Comments to language quality:
The sentences shift between present and past tenses. For example, in the sentence “In LS-SCLC, combining etoposide with CCRT is standard practice. This combination enhanced local tumor control and improved survival rates by exploiting the radiosensitizing properties of etoposide [52]. The EP regimen is typically paired with thoracic radiotherapy, beginning either during or immediately after the first cycle of chemotherapy. Radiation is commonly administered at a dose of 45 gray (Gy) over 30 sessions or, alternatively, twice-daily to enhance local control." shifts between present ("is standard practice") and past ("enhanced local tumor control"). It would be better to maintain consistency. The proper sentence would be: “In LS-SCLC, combining etoposide with CCRT is standard practice. This combination enhances local tumor control and improves survival rates by exploiting the radiosensitizing properties of etoposide [52]. The EP regimen is typically paired with thoracic radiotherapy, starting either during or immediately after the first cycle of chemotherapy [56]. Radiation is typically administered at a dose of 45 gray (Gy) over 30 sessions or, alternatively, twice daily to enhance local control [57]”
Author Response
- Point 1: I suggest rewriting the title to: “Etoposide as a Key Therapeutic Agent in Lung Cancer: Mechanisms, Efficacy, and Emerging Strategies” or “The Role of Etoposide in Lung Cancer Treatment: A Detailed Review of Current and Future Strategies” better emphasizing that the manuscript focuses on etoposide.
- Response 1: Thank you very much for your excellent suggestion. We have decided to change the title, as the first title you suggested best reflects the content of this review paper. “Etoposide as a Key Therapeutic Agent in Lung Cancer: Mechanisms, Efficacy, and Emerging Strategies”.
- Point 2: In Section 2, the authors should explore whether etoposide can induce various forms of cell death beyond apoptosis, including necrosis, necroptosis, ferroptosis, pyroptosis, parthanatos, entotic cell death, NETotic cell death, lysosome-dependent cell death, autophagy-dependent cell death, and immunogenic cell death. A comprehensive understanding of these complementary cell death pathways is crucial, as it may reveal alternative mechanisms by which etoposide exerts its cytotoxic effects and contribute to a more complete characterization of its therapeutic potential and side effects.
- Response 2: Thank you for your comments. Based on your comments, we have added various forms of cell death to Section 2 of the manuscript, “Mechanism of Action of Etoposide.”
- Point 3: Authors should include a section discussing the mechanisms of cellular resistance to etoposide and the strategies employed to overcome this resistance. Etoposide resistance can significantly impact treatment efficacy, leading to therapeutic failure. A comprehensive understanding of the molecular mechanisms underlying resistance—such as the role of drug efflux pumps, alterations in drug targets, enhanced DNA repair pathways, and dysregulation of apoptosis—can elucidate the reasons why certain tumors exhibit diminished sensitivity to the drug over time. This knowledge is essential for the development of strategies to overcome resistance. Additionally, such insights may enable the design of personalized treatment approaches, wherein patients receive therapies tailored to the specific resistance mechanisms present in their tumors, thereby improving therapeutic outcomes and minimizing unnecessary side effects.
- Response 3: Thank you for your advice. We have changed the title of the "5. Toxicity and Side Effects of Etoposide" section in the manuscript to "5. Toxicity, Side Effects, and Resistance Mechanisms of Etoposide" and added information on etoposide resistance.
- Point 4: Line 17: I suggest changing “breakages” into “breaks”
- Response 4: Thank you for your suggestion. We have changed "breakages" to "breaks" in the manuscript.
- Point 5: Line 91: space lacking in “TOPâ…¡facilitates”
- Response 5: Thank you for pointing out the mistake. We have corrected the original manuscript by adding a space in "TOPâ…¡ facilitates".
- Point 6: Line 120-122: sentence “Etoposide, a topoisomerase II inhibitor, plays a pivotal role in these regimens by inducing DNA damage, leading to cancer cell apoptosis” should be removed.
- Response 6: Thank you for your comment. We have removed the sentence “Etoposide, a topoisomerase II inhibitor, plays a pivotal role in these regimens by inducing DNA damage, leading to cancer cell apoptosis” from the manuscript.
- Point 7: Line 127: lack of full names for abbreviations: “such as PD-1/PD-L1 inhibitor”
- Response 7: Thank you for pointing out the mistake. We have added the full names to the manuscript.
- Point 8: Line 128: The sentence is relatively clear but could benefit from some refinement for greater precision and flow e.g. "The introduction of immunotherapies, particularly immune checkpoint inhibitors such as pembrolizumab and nivolumab, has significantly expanded treatment options for advanced lung cancer. These agents block inhibitory signals in the immune system, thereby boosting antitumor activity."
- Response 8: Thank you for your comment. We have revised the sentence you mentioned in the manuscript to better convey the meaning.
- Point 9: Line 139 and 232: “Etoposide, a potent topoisomerase II inhibitor/Etoposide, a potent chemotherapeutic agent that inhibits topoisomerase II ”information is mentioned for the third time in the manuscript and should be removed.
- Response 9: Thank you for your feedback. We have removed the repeated phrase "Etoposide, a potent topoisomerase II inhibitor/Etoposide, a potent chemotherapeutic agent that inhibits topoisomerase II" from the manuscript.
- Point 10: Line 143: “this can help…” can be replace with “thus help…”
- Response 10: Thank you for your comment. We have revised "This can help..." to "thus help..." in the manuscript.
- Point 11: Line 153, 173, 190: remove the bold.
- Response 11: Thank you for pointing out our mistake. We have removed the bold text from the manuscript.
- Point 12: Line 126: “immune checkpoint inhibitors” appear the first time in the text and is later shortened in line 177. Please use abbreviation (ICIs) the first time it appears in the text.
- Response 12: Thank you for pointing out our mistake. We used the abbreviation when the text "immune checkpoint inhibitor" first appeared in the manuscript and then abbreviated it thereafter.
- Point 13: Line 205: Include full name for CDK abbreviation.
- Response 13: Thank you for your comment. We have included the full name of the CDK in the manuscript.
- Point 14: Line 245: I suggest replacing “bolster” with “boost” or “enhance”
- Response 14: Thank you for your comment. We replaced "bolster" with "enhance " in the manuscript.
- Point 15: Comments to language quality:
The sentences shift between present and past tenses. For example, in the sentence “In LS-SCLC, combining etoposide with CCRT is standard practice. This combination enhanced local tumor control and improved survival rates by exploiting the radiosensitizing properties of etoposide [52]. The EP regimen is typically paired with thoracic radiotherapy, beginning either during or immediately after the first cycle of chemotherapy. Radiation is commonly administered at a dose of 45 gray (Gy) over 30 sessions or, alternatively, twice-daily to enhance local control." shifts between present ("is standard practice") and past ("enhanced local tumor control"). It would be better to maintain consistency. The proper sentence would be: “In LS-SCLC, combining et oposide with CCRT is standard practice. This combination enhances local tumor control and improves survival rates by exploiting the radiosensitizing properties of etoposide [52]. The EP regimen is typically paired with thoracic radiotherapy, starting either during or immediately after the first cycle of chemotherapy [56]. Radiation is typically administered at a dose of 45 gray (Gy) over 30 sessions or, alternatively, twice daily to enhance local control [57]”
- Response 15: Thank you very much for your advice. We have revised the sentences in the manuscript to ensure consistency without mixing present and past.

Reviewer 2 Report
Comments and Suggestions for Authors
The review article entitled "Topoisomerase II Inhibitors in Lung Cancer: A Comprehensive Review of Etoposide's Role and Emerging Therapeutic Strategies” describes the clinical application of etoposide as anticancer agent. It includes comprehensive information about the search’s point but there are some important points should be considered before accepting for publication:
1-High level of similarity 27%. Authors need to rephrase the statements to lower this level
2-Methodology part: This review is missing the methodology part. Please add this part to cover the following areas:
- Web sites used for searching process
-Search terms
- Time frame which is covered by this article
- Limitation of the searching process if there is any.
3-Authors referred to an essential medicine list that was produced from Etoposide, however they didn’t explain these medications. Write a short paragraph explaining these medication types, names, uses…. etc.
4-Write a future perspective part to describe your outlook or expectation of what might happen in the future regarding clinical applications of etoposide.
Author Response
- Point 1: High level of similarity 27%. Authors need to rephrase the statements to lower this level.
- Response 1: Thank you for your comments. We have tried to rephrase the statements as much as we can. However, due to the nature of a review paper, it was challenging to further reduce the similarity of the manuscript as we needed to use key terms directly from the references. We kindly ask for your understanding, considering that all the authors are non-native English researchers.
- Point 2: Methodology part: This review is missing the methodology part. Please add this part to cover the following areas:
- Web sites used for searching process
- Search terms
- Time frame which is covered by this article
- Limitation of the searching process if there is any.
- Response 2: Thank you for your advice. We have added a methodology section to the introduction section of the manuscript.
- Point 3: Authors referred to an essential medicine list that was produced from Etoposide, however they didn’t explain these medications. Write a short paragraph explaining these medication types, names, uses…. etc.
- Response 3: Thank you for your suggestion. We have added the World Health Organization (WHO) description of essential medicines to the "Mechanism of Action of Etoposide" section of the manuscript.
- Point 4: Write a future perspective part to describe your outlook or expectation of what might happen in the future regarding clinical applications of etoposide.
- Response 4: Thank you for your comments. We have added a future perspective section to the conclusion section of the manuscript, which describes the future prospects or expectations regarding the clinical application of etoposide.

Round 2
Reviewer 2 Report
Comments and Suggestions for Authors
The authors addressed all the comments, and I approve the manuscript for publication.